# Bioequivalence of Oral Drug Products in the Healthy and Special Populations: Assessment and Prediction Using a Newly Developed In Vitro System “BE Checker”

**DOI:** 10.3390/pharmaceutics13081136

**Published:** 2021-07-26

**Authors:** Takato Masada, Toshihide Takagi, Keiko Minami, Makoto Kataoka, Ken-ichi Izutsu, Kazuki Matsui, Shinji Yamashita

**Affiliations:** 1Faculty of Pharmaceutical Sciences, Setsunan University, Osaka 573-0101, Japan; 18d402mt@edu.setsunan.ac.jp (T.M.); toshihide.takagi@setsunan.ac.jp (T.T.); keiko.minami@pharm.setsunan.ac.jp (K.M.); makoto@pharm.setsunan.ac.jp (M.K.); 2National Institute of Health Sciences, Kanagawa 210-9501, Japan; izutsu@nihs.go.jp; 3Sawai Pharmaceutical Co. Ltd., Osaka 532-0003, Japan; kazuki.matsui@sawai.co.jp

**Keywords:** oral drug product, bioequivalence, special population, gastrointestinal physiology, dissolution, permeation, gastric emptying

## Abstract

In order to assess and predict the bioequivalence (BE) of oral drug products, a new in vitro system “BE checker” was developed, which reproduced the environmental changes in the gastrointestinal (GI) tract by changing the pH, composition, and volume of the medium in a single chamber. The dissolution and membrane permeation profiles of drugs from marketed products were observed in the BE checker under various conditions reflecting the inter-patient variations of the GI physiology. As variable factors, initial gastric pH, gastric emptying time, and GI agitation strength were varied in vitro. Dipyridamole, a basic drug, showed rapid and supersaturated dissolution when the paddle speed in the donor chamber was 200 rpm, which corresponds to the high agitation strength in the stomach. In contrast, supersaturated dissolution disappeared, and the permeated amount decreased under the conditions with a slow paddle speed (100 and 50 rpm) and short gastric emptying time (10 min). In those conditions, disintegration of the formulation was delayed, and the subsequent dissolution of dipyridamole was not completed before the fluid pH was changed to neutral. Similar results were obtained when the initial gastric pH was increased to 3.0, 5.0, and 6.5. To investigate that those factors also affect the BE of oral drug products, dissolution and permeation of naftopidil from its ordinary and orally disintegrating (OD) tablets were observed in the BE checker. Both products showed the similar dissolution profiles when the paddle speed and gastric emptying time were set to 100 rpm and 10 or 20 min, respectively. However, at a low paddle speed (50 rpm), the dissolution of naftopidil from ordinary tablets was slower than that from the OD tablets, and the permeation profiles became dissimilar. These results indicated the possibility of the bioinequivalence of some oral formulations in special patients whose GI physiologies are different from those in the healthy subjects. The BE checker can be a highly capable in vitro tool to assess the BE of oral drug products in various populations.

## 1. Introduction

The bioequivalence (BE) study of oral dosage forms is performed to assure therapeutic equivalence of two drug products containing the same active pharmaceutical ingredient (API), by comparing the rate and extent of drug absorption after oral administration [1,2,3]. After in vitro tests for drug release/dissolution, human BE study is usually performed using healthy adult volunteers. Subjects having undesirable diseases or who are taking concomitant medicines are excluded from these studies according to the exclusion criteria. This is to avoid the risk of adverse events caused by the test or reference products, and the study design with healthy subjects is rational. However, once the drug products are approved and put on the market, various patients having diverse chronic disorders, health conditions, or lifestyles may take them. Such patients often use generic drug products as a switch from original products. Switched products will show the same efficacy and safety as the original ones for most patients because they are proved to be BE. However, that may not be always the case for certain populations. As a typical example, poorly soluble basic drugs are known to be dissolved in the stomach at a low pH and are then transferred to the small intestine as a solution to be absorbed rapidly. In patients with achlorhydria or with concomitant use of proton pump inhibitors (PPIs), the dissolution of basic drugs in the stomach will be inhibited, and the absorption from the small intestine will be decreased. In those patients, the rate and extent of drug absorption becomes strongly dependent on the formulation design, and some BE products may show different profiles of absorption that could not be detected in a BE study with healthy subjects. Other factors of GI physiology such as gastric emptying rate, agitation strength, GI fluid compositions (including bile concentrations), and volumes can cause the inequivalent performance of BE products in the special population. Although the concept of “BE in the special population” is widely recognized as a critical issue when using a generic drug product, clear rules are not yet defined because of the lack of clinical evidence and the methodology to detect it.

Due to the less biorelevant conditions that are currently used in in vitro drug dissolution tests defined by the Pharmacopeia in each country (typically a paddle or a basket in 900 mL vessels), various kinds of in vivo predictive in vitro methods for investigating drug dissolution in the GI tract have been developed. Several types of multiple compartmental GI transfer systems have been developed so far, in which two or three dissolution vessels are connected by peristatic pumps for analyzing the drug dissolution in each part of the GI tract [4,5,6,7,8,9,10,11]. As one of those systems, a multicompartmental in vitro dissolution apparatus, the gastrointestinal simulator (GIS), was shown to be able to detect the effect of gastric pH on the dissolution profile of dipyridamole, a BCS class II weak base drug, which corresponded well to the clinically observed drug–drug interaction with famotidine [12]. An additional function of drug partitioning to the octanol phase was also introduced to the GIS to observe the process of membrane permeation [13]. Further, as a more complex system, the TNO gastro-intestinal model is designed to realistically simulate conditions in the GI tract. Although such systems are successful for simulating the performance of drug formulations in the GI tract, applications to the test these drugs in various GI conditions representing those in particular patients are limited because of their complex and large-scale systems, thus low throughput and difficulty in setting up the system for variable GI conditions.

In a previous study [14], we have developed an in vitro system (named the Stomach-to-Intestine Fluid Changing (SIFC) system) which can reproduce the temporal changes in the pH, composition, and volume of the GI fluid in a single vessel equipped with one syringe pump. In the SIFC system, the fasted state gastric simulated fluid (FaSSGF) was used as an initial fluid in the vessel to which drugs could be applied as a powder or formulation. The concentrated fasted state simulated intestinal fluid (pre-FaSSIF) was then infused to the vessel at a constant speed to change the entirety of the fluid to FaSSIF (pH 6.5). For the applied drug or drug formulation, this process corresponded to the transit from the stomach to the small intestine after oral administration. One of the most advantageous points of this system is that it is possible to respond to a variety of GI conditions by simply changing the fluid pH or composition, and infusion rate of the GI system. Using this system, the effects of the initial gastric pH and dose strength in the formulation of telmisartan were observed to simulate dose-dependent intra-subject variability in the plasma drug concentration in human BE studies.

In this study, based on the previously developed SIFC system, a more biorelevant in vitro system (in vitro bioequivalence checking system (BE checker)) was newly designed to predict the BE of oral drug products in healthy and specific populations. Not only the dissolution but also the membrane permeation of drugs from marketed products were observed. As they are the GI physiological factors that influence the drug release/dissolution profile, effects of the initial gastric pH, gastric emptying rate, and the agitation strength in the GI tract were investigated using the BE checker to demonstrate the risk of bioinequivalence in special populations.

## 2. Materials and Methods

### 2.1. Materials

Hank’s balanced salt solution (HBSS) was obtained from Gibco Laboratories (Lenexa, KS, USA). Egg-phosphatidylcholine (lecithin) was obtained from FUJIFILM Wako Pure Chemical (Osaka, Japan). Sodium taurocholate (NaTC) was obtained from Nacalai Tesque (Kyoto, Japan). Seloken^®^ tablet 20 mg and Persantin^®^ tablets 25 mg were obtained from AstraZeneca (Osaka, Japan) and Nippon Boehringer Ingelheim (Tokyo, Japan), respectively. Flivas^®^ tablets 25 mg and Flivas^®^ orally disintegrating (OD) tablets 25 mg were obtained from Asahi Kasei Pharma (Tokyo, Japan). All other reagents used were of the highest purity.

### 2.2. Design of In Vitro Bioequivalence Checking System for Oral Drug Product (the BE Checker)

The BE checker developed in this study is illustrated in Figure 1. A hydrophilic filter with a pore size of 0.22 µm (Durapore membrane filter, MERCK Millipore, Burlington, MA, USA) was mounted between the donor and receiver chambers. The effective surface area of the filter was 6.60 cm^2^. Both chambers were made of acrylic plastic. The volume of the donor chamber was 40 mL in the bottom, which corresponds to the stomach, and the model had a total volume of 100 mL. The volume of the receiver chamber was set to 30 mL.

The compositions of FaSSGF and FaSSIF used in this study are summarized in Table 1. At the beginning of the experiment, the bottom of the donor chamber was filled with 40 mL of FaSSGF. After administration of the tablet, 1.67-times concentrated FaSSIF (pre-FaSSIF) was infused into the donor chamber to shift the donor fluid to FaSSIF. Since HBSS contains 19.45 mM d-glucose, the final concentration of the d-glucose in the FaSSIF was 25 mM.

The pH of the FaSSGF was adjusted to 1.6, 3.0, 5.0, or 6.5 using the diluted HCl and/or NaOH solution. Pre-FaSSIF was prepared to finally construct 100 mL of FaSSIF (pH 6.5) after its addition to 40 mL of FaSSGF.

The receiver chamber was empty at the beginning, and octanol was then infused into the receiver side at a rate to keep the height of the fluid surface the same as the donor side.

### 2.3. Simultaneous Evaluation of Disintegration/Dissolution and Permeation of the Drugs from the Tablets Administered to the BE Checker

Experimental conditions designed to represent the GI physiology in the fasted state are shown in Figure 2. A tablet was administered to 40 mL FaSSGF in the bottom of the donor chamber. After agitation for one minute as a disintegration/dissolution phase, 60 mL pre-FaSSIF was infused into the donor chamber at a constant rate for 10 or 20 min using a syringe pump as gastric emptying phase. At the same time, water-saturated octanol was infused into the receiver side. After infusion, dissolution and permeation of the drug in the small intestine were evaluated. Samples were taken from each side of the chamber at the specified time. Donor samples were immediately filtered through a membrane filter (Millex®-LH, pore size: 0.45 µm, Millipore, MERCK Millipore, Burlington, MA, USA) to remove any undissolved and precipitated drug. The volume of each side was maintained by adding fresh fluid after sampling. Samples were stored at −80 °C until analysis.

### 2.4. Analytical Methods

Drug concentrations in the samples were determined using an ultra-performance liquid chromatography (UPLC) system (ACQUITY^®^ UPLC, Waters, MA, USA) equipped with a tandem mass spectrometer (ACQUITY TQD, Waters, MA, USA). In metoprolol and dipyridamole, the donor and receiver samples were diluted with 0.1% formic acid/acetonitrile (50:50) and water/acetonitrile (20:80), respectively. In the case of naftopidil, the donor and receiver samples were diluted with 0.1% formic acid/methanol (50:50) and water/methanol (20:80), respectively. A reversed-phase Waters Acquity UPLC BEH C18 analytical column (50 mm × 2.1 mm, 1.7 µm particle size) was used with a mobile phase consisting of 0.1% (*v/v*) formic acid (solvent A) and acetonitrile containing 0.1% (*v/v*) formic acid (solvent B) with a gradient time period. In the case of naftopidil, 0.1% (*v/v*) formic acid was used as solvent A, and methanol was used as solvent B. The initial mobile phase was 98% solvent A and 2% solvent B pumped at a flow rate of 0.3 mL/minutes. Between 0 and 1.0 min, the percentage of solvent B increased linearly to 95%, which was then maintained for 1.0 min. Between 2.01 and 2.5 min, the percentage of solvent B decreased linearly to 2%. This condition was maintained until 3 min had pased, at which time the next sample was injected into the UPLC system. All samples were injected at a volume of 5 μL into the UPLC system. The ionization conditions for the analysis of metoprolol were as follows: electrospray ionization condition, positive mode; source temperature, 150 °C; desolvation temperature, 400 °C; cone voltage, 10 V; and collision energy, 50 V. Precursor and production ion (*m/z*) were 267.67 and 77.05, respectively. The ionization conditions for the analysis of dipyridamole were as follows: electrospray ionization condition, positive mode; source temperature, 150 °C; desolvation temperature, 400 °C; cone voltage, 30 V; and collision energy, 48 V. The precursor and production ion (*m/z*) were 505.33 and 385.48, respectively. The ionization conditions for the analysis of naftopidil were as follows: electrospray ionization condition, positive mode; source temperature, 150 °C; desolvation temperature, 400 °C; cone voltage, 40 V; and collision energy, 30 V. Precursor and production ion (*m/z*) were 393.39 and 190.09, respectively.

## 3. Results

### 3.1. pH-Shift Profiles in the Donor Chamber of the BE Checker

A total of four types of pH-shift profiles from the donor side reproduced in this study are shown in Figure 3. The pH of the initial FaSSGF was set to 1.6, 3.0, or 5.0 to mimic the gastric pH of healthy or achlorhydric patients. In all protocols, the fluid pH increased gradually with the infusion of pre-FaSSIF, and the profiles of the pH shift were smooth in all pH ranges. Since all fluids contained Na_2_HPO_4_ and citric acid as buffering agents, this profile corresponded to that of the gradual pH change protocol in our previous report [14]. The final pH after completing the pre-FaSSIF infusion was close to 6.5 in all experimental conditions.

### 3.2. Effect of Paddle Speed on the Dissolution and Permeation of Metoprolol from Seloken^®^ Tablets

Paddle speed was set to 50, 100, or 200 rpm in the BE checker to observe the effect of agitation strength on the dissolution of metoprolol from its marketed product. A Seloken^®^ tablet 20 mg was used in this study since it was demonstrated that metoprolol dissolves completely in the in vitro dissolution study of the Japanese Pharmacopeia (JP), and its dissolution is not influenced by the fluid pH (paddle apparatus with pH 1.2, 4.0, and 6.8 medium) [15]. Initial gastric pH and pre-FaSSIF infusion time were fixed to pH 1.6 and 10 min, respectively.

As shown in Figure 4, the dissolution of metoprolol from the Seloken^®^ tablet became faster as the paddle speed increased and at 200 rpm, and more than 80% of the metoprolol was dissolved in the donor chamber within 30 min. This profile of metoprolol dissolution is similar to that of the one from the dissolution test in the JP (paddle rotation speed = 50 rpm, 20 mg in 900 mL) [15]. The dissolution was slowed at 100 and 50 rpm and reached to 80% within 45 min at 100 rpm and 60 min at 50 rpm, respectively. Since the shape of apparatus and the dose/fluid volume ratio were different between the dissolution test in the JP and the BE checker, it is not possible to directly compare the correlation between the paddle speed and drug dissolution profiles in both studies. However, these experimental conditions, 50 to 200 rpm of paddle speed in the BE checker, were considered to cover a reasonable range of agitation strength in the human GI tract for testing the dissolution of drugs, although it is difficult to define which speed is average in the healthy population.

The effect of paddle speed on the permeation profile of metoprolol was not clearly observed despite its pronounced effect on the early phase-profile of the dissolution. Since the filter was not soaked with the donor and receiver fluids at the beginning and because the fluid level gradually raised to initiate the permeation of the drug, early-phase differences in the dissolution profile may not be completely reflected in the permeation profile. This is a similar situation to in vivo experiments because the drug dissolution profile in the stomach just after drug administration may not profoundly affect the absorption profile of the intestine.

### 3.3. Effect of Paddle Speed, Pre-FaSSIF Infusion Time and Initial Fluid pH on the Dissolution and Permeation of Dipyridamole from Persantin^®^ Tablets in the BE Checker

Dipyridamole is a weak basic drug and is highly soluble to water at a low pH (>100,000 µg/mL at pH < 2.0 and 7000 µg/mL at pH 3) but is less soluble at a neutral pH (6 µg/mL at pH 6.5). The dipyridamole dissolution profile from its marketed product, Persantin^®^ tablets 25 mg, using the paddle apparatus was reported to be strongly affected by medium pH in the dissolution study. Dipyridamole dissolved almost completely within 10 min at pH 1.2 but less than 20% after 360 min at pH 6.5 [15].

Figure 5A shows the dissolution profiles of dipyridamole from the Persantin^®^ tablet in the BE checker at the paddle speeds of 50, 100, or 200 rpm. The initial gastric pH and pre-FaSSIF infusion time were fixed to pH 1.6 and 10 min, respectively. At the 200 rpm paddle speed, dipyridamole was dissolved completely within 5 min, and the dissolved amount in the donor chamber then decreased gradually. Because the infusion of pre-FaSSIF was completed and the fluid pH was changed to 6.5 by the 10 min point, this time-profile of the dissolved amount represented a precipitation of dipyridamole from the supersaturated solution caused by the pH shift from 1.6 to 6.5. In contrast, at 100 or 50 rpm of paddle speed, supersaturation was not observed, and the dissolved amount of dipyridamole was kept at less than 25% throughout the experiment. Permeation of dipyridamole to the receiver chamber in Figure 5B well reflected the difference in the time-profile of dissolved amount.

The pre-FaSSIF infusion time was then prolonged to 20 min, and the same experiments were performed. As shown in Figure 6, initial supersaturation and subsequent precipitation were not only observed at 200 rpm, but also at 100 and 50 rpm of paddle speed. The dissolved and permeated amounts were about the same at 100 and 200 rpm, while those at 50 rpm were lower during the initial 30 min of the experiment. Since the pre-FaSSIF infusion time in the BE checker in vitro corresponds to the time needed to transfer the drug from the stomach to the small intestine in vivo, the results in Figure 5 and Figure 6 imply that the combination of the agitation strength and the gastric emptying time is possible to cause the variety of dissolution profiles and the absorption of dipyridamole in vivo.

In Figure 7, as another factor to influence the dissolution profile of dipyridamole, the initial pH of FaSSGF was varied from 1.6 to 6.5. For other experimental conditions, pre-FaSSIF infusion time and paddle speed were fixed to 20 min and 100 rpm, respectively.

As already shown in Figure 5, a supersaturated dissolution and higher permeation of dipyridamole were observed when the initial FaSSGF pH was set to 1.6, whereas at the higher initial pH conditions, this supersaturation profile disappeared, and the dissolved and permeated amounts of dipyridamole decreased as the initial pH increased. From the pH-solubility profile of dipyridamole, this result seems reasonable and suggests the possibility of lower absorption in patients with high gastric pH.

### 3.4. Application of the BE Checker for the Assessment of BE between Flivas^®^ Tablets and Flivas^®^ OD Tablets

To investigate whether the difference in the experimental conditions in the BE checker reflect whether the variable GI physiology causes the inequivalent performances of two BE products, ordinary and OD tablets of naftopidil (Flivas^®^ tablets and Flivas^®^ OD tablets) were used in the experiment. Flivas^®^ OD tablets have been proved to be BE with Flivas^®^ tablets in human BE studies and are approved for clinical use. Naftopidil is a weak base drug with a pKa of 3.7 and 6.7 and shows quite low solubility at a neutral pH (<0.1 µg/mL at pH > 5.0).

As shown in Figure 8, regardless of the pre-FaSSIF infusion time, the dissolution of naftopidil from the OD tablets was significantly slower than that from the ordinary tablets, and a clear peak of supersaturation was not observed. Accordingly, the permeation of naftopidil from the OD tablets was slower during the initial 30 min. Since the OD tablets are designed to disintegrate quickly in the oral cavity after administration, even without a concomitant ingestion of water, the results in Figure 8 were completely different from what we expected. To understand the reason why, JP1 solution (pH 1.2), FaSSGF (pH 1.6) or distilled water was dropped onto the OD tablet in a Petri dish, and the disintegration process was observed. Figure 9 shows the pictures at 1 and 3 min after the addition of each solution.

When JP1 or FaSSGF was added, a gel-like layer formed on the surface of the OD tablet, which disturbed further disintegration, while the addition of distilled water quickly progressed the disintegration. It was reported that the solubility of naftopidil in JP1 solution (pH 1.2, prepared by HCl solution) is lower than that in phosphate buffer at pH 2.0 (0.05 mol/L) because the solubility of naftopidil in JP1 is limited by a solubility equilibrium (product) with chloride ions [16]. The formation of the gel-like layer shown in Figure 9 might be due to the aggregation of quickly dissolving and precipitating naftopidil (as a complex with chloride ions) at the surface of the OD tablet. In the in vivo situation for drug administration, if the OD tablets are directly administered into the stomach, the same phenomenon will occur. However, since the OD tablets are disintegrated and dispersed quickly in the oral cavity before reaching to the stomach, the delay in the dissolution and absorption was not observed in the human BE study in vivo [16]. This is even true when the OD tablets were taken without water because the OD tablets can be disintegrated by saliva.

To reflect the situation of in vivo oral drug administration, the process of tablet disintegration in water was introduced to the experimental protocol of the BE checker. In the new protocol, first, the bottom of the donor chamber was filled with 37.5 mL of distilled water. After 1-min of incubation with the tablet, 0.25 mL of concentrated FaSSGF was added to set the donor fluid volume as 40 mL FaSSGF (pH 1.6), assuming the orally administered tablet will reach the stomach in 1 min.

By introducing this disintegration time to the experimental protocol, dissolution of naftopidil from the OD tablets became faster and showed similar dissolution and permeation profiles to those from the ordinary tablets (Figure 10). In this experiment, a new protocol with 1-min disintegration in water was applied both for the ordinary and OD tablets. The longer infusion time of pre-FaSSIF (20 min) increased the dissolved and permeated amount of naftopidil, whereas the dissolution and permeation profiles were almost the same between the ordinary and the OD tablets.

Using the same protocol including 1-min disintegration time, the effects of agitation strength on the dissolution and permeation of naftopidil were observed by changing the paddle speed to 50 rpm (Figure 11). As seen in Figure 11, the dissolved and permeated amounts of naftopidil decreased significantly from both the ordinary and the OD tablets, compared to those in Figure 10 at 100 rpm of paddle speed.

However, since the effect of decreased paddle speed was more pronounced on the ordinary tablet, clear differences in dissolution and permeation profiles were observed between the ordinary and OD tablets, at both pre-FaSSIF infusion times of 10 and 20 min.

To quantitatively assess the similarity level between the ordinary and OD tablets, the f_2_ value was calculated as a similarity factor for the naftopidil permeation profiles from two products in the BE checker using the following equation
(1)$$f_{2}=50\times \log\left[\frac{100}{\sqrt{1+\frac{\sum_{i=1}^{n}{\left(\mathrm{Ti}-\mathrm{Ri}\right)}^{2}}{n}}}\right]$$
where Ti and Ri indicate the permeated amount of naftopidil (% of dose) from the ordinary tablets and the OD tablets, respectively, and *n* indicates the number of measurement points to be compared. In the calculation, f_2_ was evaluated at four time points, 1/4Ta, 1/2Ta, 3/4Ta, and Ta, where the permeated amount from the ordinary tablet at 120 min was set as 100%, and Ta was the time point of about 85%. Table 2 represented the similarity factor (f_2_ value) obtained from the results in Figure 8, Figure 10, and Figure 11. If the border of the similarity was set to 50, it was could be concluded that both tablets gave similar profiles of naftopidil permeation at the 100 rpm paddle speed when the protocol with disintegration in water was applied, whereas at 50 rpm, a similarity in the permeation profile was not obtained. These results might imply that the ordinary and OD naftopidil tablets are BE in subjects with strong (or normal) stomach agitation strength, but there is a risk of being bioinequivalence in subjects whose agitation strength is weak.

## 4. Discussion

Although various in vitro systems have been developed and used to assess the bioavailability and/or bioequivalence of oral drug products, starting with a simple dissolution test, it is fair to say that no in vitro systems can perfectly mimic these highly complex in vivo drug absorption processes. For all systems, there are some pros and cons. With those as a basic premise, we have designed a new in vitro system, the BE checker, for testing the BE of oral drug products with the following questions in mind;

✓How is the system biorelevant?✓How can the various physiological conditions in the GI tract be reflected?✓What kind of data can be obtained from evaluating BE?✓What is the throughput and cost-performance?

As a biorelevant system, it is essential to expose drug products to different environments in the GI tract, to those in the stomach first, and then to those in the small intestine. Multiple compartmental GI transfer systems reproduced these processes by connecting several vessels filled with biorelevant fluids [4,6,7,8,9,10,12,13]. Small particles or dissolved drugs in the fluids were transferred from one vessel to the next using a peristatic pump to mimic a GI transit in vivo. Although such compartmental systems are successful in observing the disintegration/dissolution of drug formulations in each part of the GI tract, systems are relatively large and complicated. In addition, the peristatic pump sometimes fails to adequately transfer the drug particles that are not dissolved or yielded by precipitation.

In the BE checker, the GI transit of drug formulations or drugs was embodied by changing the surrounding environments while the drugs were kept in the same vessel. This is the most important and advantageous feature of this system, which was adapted from the previously developed SIFC system [14]. This concept is noted as “environmental changes in the GI tract from the stomach to the small intestine are reproduced in one chamber by gradually changing the fluid pH, composition and volume without transferring drugs or drug formulations”. Only one donor chamber equipped with one syringe pump is needed to achieve this process in the BE checker.

Another point to be addressed is that a receiver chamber was added to the system, which, when equipped with another pump to infuse octanol as a receiver fluid, can observe the membrane permeation process of the dissolved drug. We have already developed a small-scale, side-by-side chamber system and dissolution/permeation (D/P) system to investigate the oral absorption of drugs, mainly the fraction/dose absorbed from the small intestine (Fa). In the D/P system, a Caco-2 cell monolayer was used as a model membrane of the human intestine, and dissolved and permeated amounts (during the defined period) were measured after the addition of the drug to the donor chamber as a powder. A permeated amount in the D/P system was then converted to the Fa found in humans by using a standard curve obtained in advance using various marketed drugs. The D/P system is now used to screen the oral absorbability of APIs as well as to check the effect of food intake or different formulations on it [17,18,19]. In the BE checker, instead of a Caco-2 cell monolayer, a hydrophilic filter was mounted between the chambers to prevent the mixing of donor and receiver fluids. As a receiver fluid, octanol was used because the partition of octanol is known as a good surrogate for drug permeation across the intestinal membrane. In an in vitro bi-phasic system, octanol is used as an absorptive compartment [11]. Shi et al. have measured the time-profile of a drug amount partitioned to octanol from the dissolution medium in a bi-phasic system and reported its high possibility of assessing the drug absorption profile in vivo [20]. In the BE checker, the height of the fluid surface on both sides is kept always the same by adjusting the octanol infusion speed. The surface area available for drug permeation gradually increases as the amount of the fluid increases. This may partially reflect the in vivo process of fluid spreading into the intestinal tract to increase the membrane surface area for drug absorption.

As with the case of a D/P system, the amount of the permeated drug (% of dose) in the BE checker was much smaller than that in vivo absorption (Fa). This is simply due to the lower permeation (absorption) clearance of the filter/octanol system in the BE checker than that in the in vivo absorption from the small intestine because of the smaller membrane surface area/volume ratio (A/V ratio) in this system. To evaluate the Fa from the data from the BE checker, the same method as that used for a D/P system is applicable in that the correlation between the permeated amount and Fa was used as a standard curve. Additionally, it is possible to convert the drug permeation profile in the BE checker to its plasma concentration profile after oral administration using in silico PK modeling. The obtained parameters such as AUC and C_max_ will then be used to predict the BE of the test formulation. This is the next project to make the BE checker more useful for BE analysis using in vitro*–*in silico–in vivo extrapolation (IVISIVE). The study is currently in progress and will soon be reported.

In this study, the BE checker was designed as a 1/4 scale model of the human GI tract in volume. In some cases, drug products containing a 1/4 dose API of maximum dose strength are commercially available. By using a 1/4 dose strength product, the result in the BE checker can be regarded as representing the performance of the product with maximal dose strength in vivo, although the ratio of other excipients might be different. This is to reduce a cost and increase the throughput of the experiment. If necessary, it is not difficult to prepare a 1/1 scale system with a similar shape and functions.

The fasted state fluid volume in the stomach immediately after the ingestion of 150 mL of water (standard volume used for human BE study in Japan) was reported as 160–200 mL [21]; thus, the initial volume of FaSSGF was set to 40 mL in the BE checker. The final volume of the donor chamber (as FaSSIF) should correspond to the volume of the small intestine. However, in this study, it was defined as 100 mL despite the fact that the fasted state intestinal volume was reported to be around 100–200 mL, even after the ingestion of 250 mL of water [22]. This is due to the absorption of water by the small intestine; thus, it was assumed that the total volume of water to which drugs are exposed to in the small intestine is larger than the reported value. To determine whether the final volume of FaSSIF in the BE checker is adequate or not to assess BE, additional studies should be done to compare the results in the BE checker with those in human BE studies for various drug products. Those studies are now on going and will be reported in the near future.

Other factors of the GI physiology relating to drug dissolution/absorption were also reflected in the experimental protocol of this study by considering the variation among patients. Among them, agitation strength in the stomach is the most difficult to be parameterized for in vitro study because in vivo strength in the stomach is not uniform, and the peristaltic movement is too complex to correctly mimic with paddle rotation. In the results using the Seloken^®^ tablet (Figure 4), a 200-rpm paddle speed in the BE checker gave a dissolution profile similar to that of metoprolol observed at 50 rpm of the paddle apparatus (in 900 mL vessel) [15]. Although a 50-rpm paddle speed is often used in dissolution tests, it does not necessarily reflect the in vivo situation properly, and in vivo agitation strength is generally considered to be weaker [23]. In this study, including the inter-subject variability, following experiments were done at three different paddle speeds, 50, 100, and 200 rpm.

Although human gastric pH is generally considered to be less than 2 in fasted condition, it was reported that a higher gastric pH (>3) was detected in about 10% of subjects, even among young healthy people [24]. In the elder population, a relatively high percentage of achlorhydric subjects whose gastric pH is almost neutral in all conditions was found in Japan [25,26,27]. To reflect these inter-subject variabilities, in this study, the initial gastric pH was varied from 1.6 to 6.5, as shown in Figure 7.

In the study with the BE checker, the infusion time of pre-FaSSIF corresponds to the gastric emptying time in vivo. In our previous report, in which the gastric emptying rate of the ingested fluid was evaluated in humans, the half-life of the fluid emptying was calculated as 3–5 min [21]. Takeuchi et al. reported that when the gastric half-life was set to 8 min, multiple compartment systems gave the most relevant dissolution profile of propranolol and metoprolol for simulating its plasma concentration profile in humans [4]. Therefore, we have employed 10 and 20 min as the pre-FaSSIF infusion times. In addition, since longer half times for gastric emptying were also presented in some reports [28,29], we have checked the dissolution and permeation profiles of dipyridamole by changing the pre-FaSSIF infusion time to 30 min as well as to 10 and 20 min. A thirty-minute infusion time gave the similar results to those obtained with a 20-min infusion time (data not shown). Subsequent studies were then performed with 10 and 20 min of pre-FaSSIF infusion.

Results in Figure 5; Figure 6 clearly indicated the importance of both the agitation strength in the stomach and the gastric emptying time to evaluate the oral absorption of dipyridamole, which dissolved rapidly in the stomach under low pH conditions and then precipitated in the small intestine. In the patients with low agitation strength and fast gastric emptying, the absorption of dipyridamole might decrease due to a less supersaturated dissolution in the stomach. This is also the same for the patients with high gastric pH, as demonstrated by the experiments with various initial FaSSGF pHs (Figure 7). Clinically, the absorption of dipyridamole is known to be inhibited by co-medication with PPIs due to elevated gastric pH [27]. The results with the BE checker corresponded well to the in vivo observations of the decreased absorption of dipyridamole in such patients.

Although the effects of GI physiological factors such as gastric agitation strength, emptying time, and fluid pH on oral drug absorption primarily depend on the physicochemical properties of APIs, if the effects are also formulation-dependent, it is possible to affect and deviatethe bioequivalency of the products in special populations. The ordinary and OD naftopidil tablets, which have been proved as BE in human BE studies, were used to check the possibility of bioinequivalence in special GI situations. As a result, in the first experiment, the OD naftopidil tablets showed slow and incomplete dissolution profiles in the BE checker compared to the ordinary tablets under standard experimental conditions (initial gastric pH was 1.6, paddle speed was 100 rpm, pre-FaSSIF infusion time was 10 or 20 min), even though the OD tablet was designed to disintegrate rapidly in the oral cavity. It was revealed that the solubility of naftopidil in a solution containing a high concentration of HCl, such as FaSSGF or JP1, is limited by a solubility equilibrium with chloride ions and, in such solutions, a gel-like layer was formed on the surface of the OD tablets due to the aggregation of the quickly dissolved and precipitated naftopidil as a complex with chloride ions (Figure 9). The effect of the gel-like layer was not observed in the in vivo BE studies (thus being proved as BE) because the OD tablets were quickly disintegrated and dispersed in the ingested water (or in the saliva) before arriving in the stomach. To mimic this situation, a 1-min disintegration time in water was added to the experimental protocol. Since this new protocol is relevant to the process of in vivo oral drug administration, subsequent studies including those with ordinary tablets were done according to it.

In the conditions where initial gastric pH was 1.6 and the paddle speed was 100 rpm, the ordinary and OD tablets showed similar dissolution and permeation profiles regardless of the pre-FaSSIF infusion time (Figure 10), suggesting the BE of these two products in the healthy population. However, this might not be true for the special patients whose gastric agitation strength is weaker. As shown in Figure 11, low paddle speed (50 rpm) caused the difference in the dissolution profile between both naftopidil products, which was less dissolved in the ordinary tablets than in the OD tablets. In the case of the OD tablets, rapid disintegration can progress the initial dissolution of naftopidil under low pH conditions, even with a low agitation strength, whereas the dissolution was highly inhibited in the ordinary tablets due to the slow disintegration in those conditions. This effect was more distinct when the pre-FaSSIF infusion time was short, although the difference was still observed at 20 min of infusion time.

The similarity level between the two formulations were quantified by calculating f_2_ values for the time-profile of the permeated drug amount (% of dose). Since an f_2_ function is usually applied to the dissolution profile in BE dissolution studies [3], the adequacy or rationality of applying this function to the drug permeation profile in the BE checker has not yet been confirmed. However, considering the features of this function [30], calculated f_2_ values can give some indications of the similarity level of the drug permeation time profiles for both products. If the border of the similarity and dissimilarity was set to 50, the result in Table 2 implies the risk of bioinequivalence in the patients with weak gastric agitation strength. It is possible to suggest that the two formulations may become bioinequivalence inpatients with gastric diseases or gastrectomy and elder or younger patients, although there is no clinical evidence to prove this at the present.

Currently, the use of in silico models and simulation technologies are accelerated and are expected to make a “virtual BE study” feasible in the near future [31]. However, even with a highly sophisticated in silico program, it is still difficult to predict the entire oral drug absorption process from various formulations under various GI conditions with a high reliability. A combination with biorelevant in vitro data is essential for the use of in silico technology in the assessment of the BE of oral drug products. As a next study with the BE checker, in vitro drug permeation data will be converted to its plasma concentration time profile in silico to predict pharmacokinetic parameters such as C_max_ and AUC in the healthy and special populations.

## 5. Conclusions

The BE checker was designed to reproduce the environmental changes in the GI tract associated with the transit of orally administered drugs from the stomach to the small intestine by changing the pH, composition, and volume of the fluid of the GI tract with a pre-set rate. This in vitro system enabled the assessment of drug dissolution/permeation profiles from oral formulations under various GI conditions, which reflected those in healthy and special populations in vivo. For ordinary and OD naftopidil tablets, different dissolution and permeation profiles were observed when the paddle speed was slowed, suggesting the possibility of bio-in-equivalent absorption in patients whose gastric agitation strength are weaker than that of healthy subjects. Although the clinical evidence pertaining to “BE in special populations” is not enough to confirm the results obtained in this study at present, the BE checker is considered to be a highly capable in vitro tool to assess the BE of oral drug products in various populations and to ensure their reliability for clinical use.

## Figures and Tables

**Figure 1 pharmaceutics-13-01136-f001:**
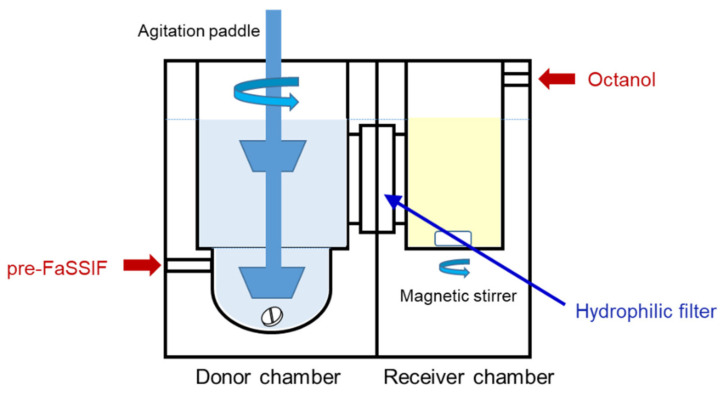
Schematic illustration of the BE checker.

**Figure 2 pharmaceutics-13-01136-f002:**
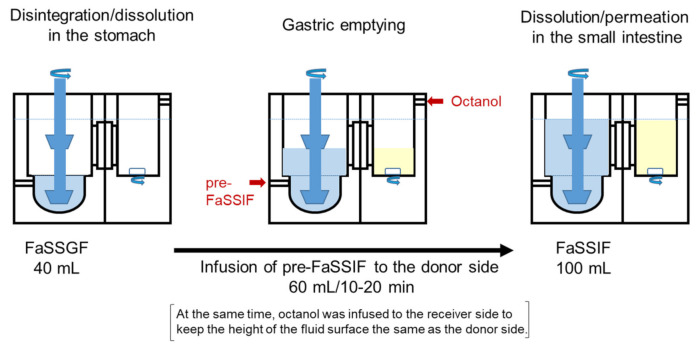
Experimental conditions represent the GI physiology in the fasted state in the BE checker.

**Figure 3 pharmaceutics-13-01136-f003:**
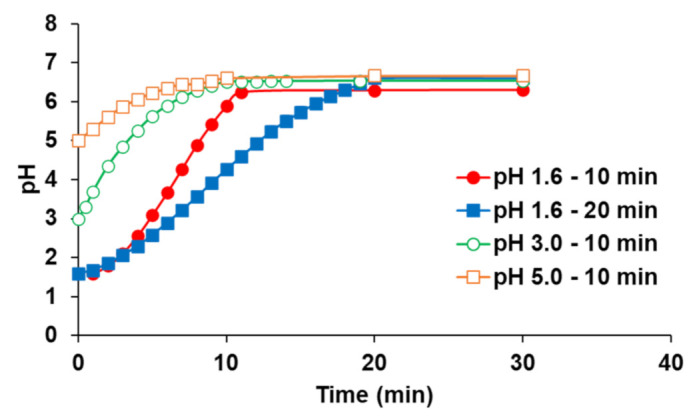
Time profile of the fluid pH in the donor chamber of the BE checker.

**Figure 4 pharmaceutics-13-01136-f004:**
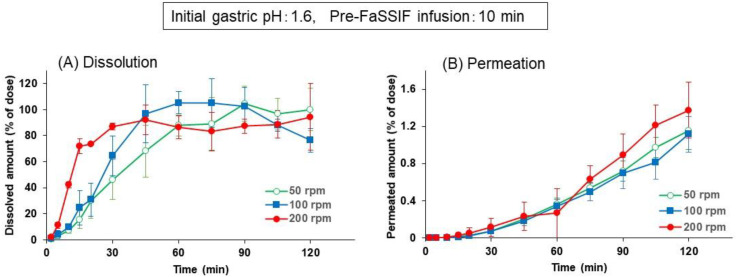
Effect of paddle speed on the dissolution (**A**) and permeation (**B**) time profiles of metoprolol from Seloken^®^ tablets 20 mg in the BE checker. Each data point represents mean ± S.D. (*n* = 3).

**Figure 5 pharmaceutics-13-01136-f005:**
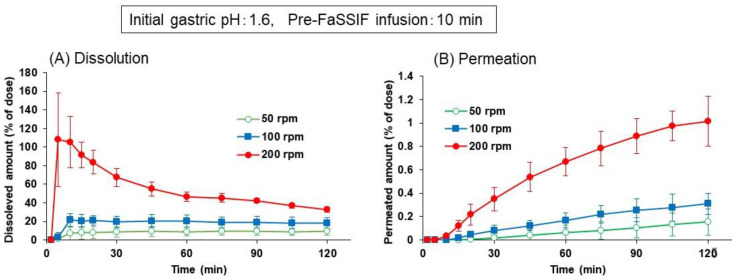
Effect of paddle speed on the dissolution (**A**) and permeation (**B**) time profiles of dipyridamole from Persantin^®^ tablets 25 mg in the BE checker. Each data point represents mean ± S.D. (*n* = 3).

**Figure 6 pharmaceutics-13-01136-f006:**
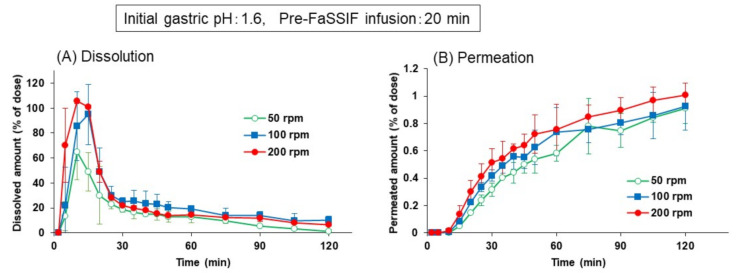
Effect of pre-FaSSIF infusion time on the dissolution (**A**) and permeation (**B**) time profiles of dipyridamole from Persantin^®^ tablets 25 mg in the BE checker. Each data point represents mean ± S.D. (*n* = 3).

**Figure 7 pharmaceutics-13-01136-f007:**
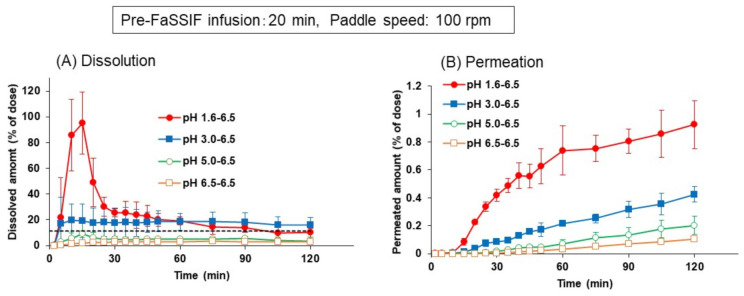
Effect of initial FaSSGF pH on the dissolution (**A**) and permeation (**B**) time profiles of dipyridamole from Persantin^®^ tablets 25 mg in the BE checker. Each data point represents mean ± S.D. (*n* = 3).

**Figure 8 pharmaceutics-13-01136-f008:**
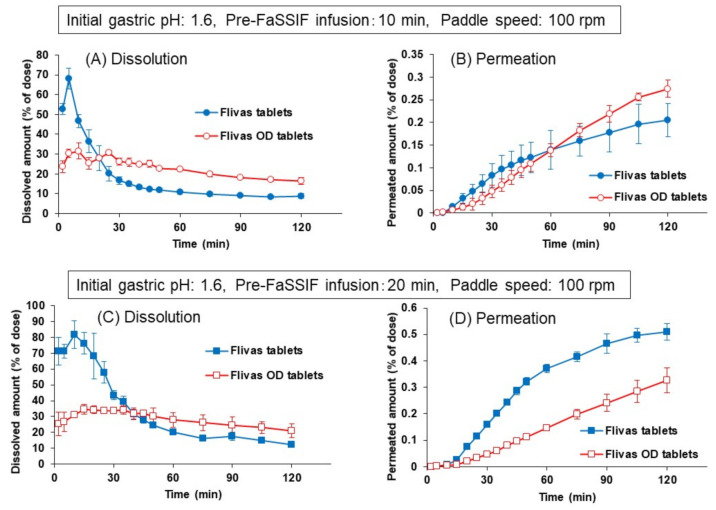
Comparison of dissolution (**A**,**C**) and permeation (**B**,**D**) of naftopidil from Flivas^®^ tablets and Flivas^®^ OD tablets in the BE checker. Pre-FaSSIF infusion time was set to 10 min (**A**,**B**) and to 20 min (**C**,**D**). Initial FaSSGF pH was 1.6, and paddle speed was 100 rpm. Each data point represents mean ± S.D. (*n* = 3).

**Figure 9 pharmaceutics-13-01136-f009:**
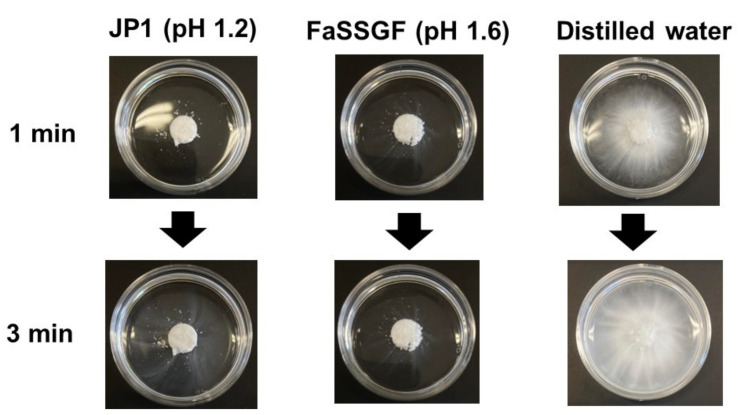
Disintegration of Flivas^®^ OD tablets after addition of JP1, FaSSGF, or distilled water onto the tablets.

**Figure 10 pharmaceutics-13-01136-f010:**
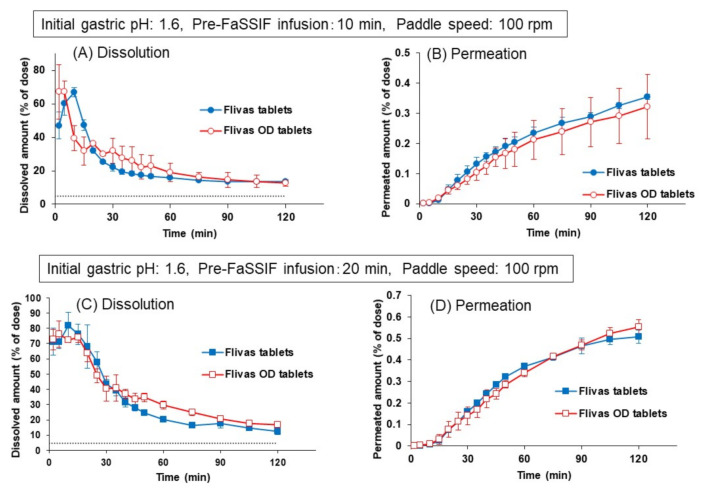
Comparison of dissolution (**A**,**C**) and permeation (**B**,**D**) of naftopidil from Flivas^®^ tablets and Flivas^®^ OD tablets in the BE checker. Experiments were performed with a new experimental protocol with 1-min disintegration time. Pre-FaSSIF infusion time was set to 10 min (**A**,**B**) or to 20 min (**C**,**D**). Initial FaSSGF pH was 1.6, and paddle speed was 100 rpm. Each data point represents mean ± S.D. (*n* = 3).

**Figure 11 pharmaceutics-13-01136-f011:**
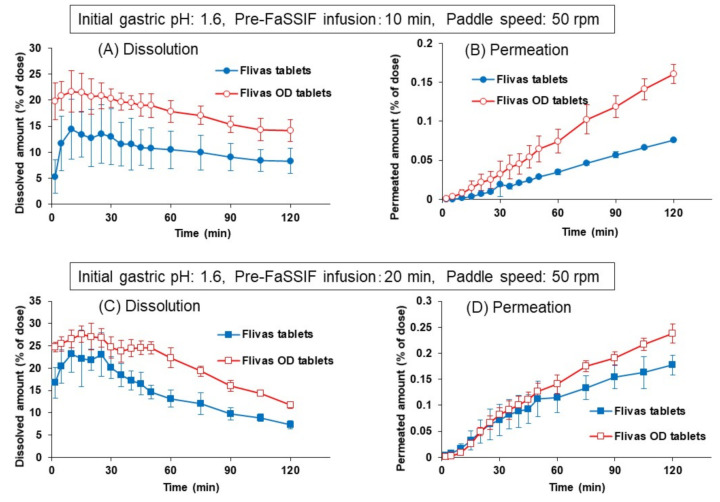
Effects of paddle speed (50 rpm) on the dissolution (**A**,**C**) and permeation (**B**,**D**) of naftopidil from Flivas^®^ tablets and Flivas^®^ OD tablets in the BE checker. Experiments were performed with a new experimental protocol with 1-min disintegration time. Pre-FaSSIF infusion time was set to 10 min (**A**,**B**) or to 20 min (**C**,**D**). Initial FaSSGF pH was 1.6. Each data point represents mean ± S.D. (*n* = 3).

**Table 1 pharmaceutics-13-01136-t001:** Compositions of the donor fluid used for the BE checker.

	FaSSGF (mM)	FaSSIF (mM)
NaTC	0.08	3
Lecithin	0.02	0.75
NaCl	34.2	13.7
Na_2_HPO_4_	20	20
Citric acid	10	10
NaHCO_3_	-	4.17
d-glucose	-	5.55
HEPES	-	10
dissolved in	Purified water	HBSS

**Table 2 pharmaceutics-13-01136-t002:** Assessment of the similarity in drug permeation profiles between Flivas^®^ tablets and Flivas^®^ OD tablets using similarity factor.

Protocols	Similarity Factor	
	Without Disintegration Time in Water	With Disintegration Time in Water
100 rpm, 10 min	31	60
100 rpm, 20 min	23	63
50 rpm, 10 min	–	34
50 rpm, 20 min	–	41

## Data Availability

All of the data described in this report is included in the publication or available in the cited references.

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
