# Peer review of "Bioequivalence of Oral Drug Products in the Healthy and Special Populations: Assessment and Prediction Using a Newly Developed In Vitro System “BE Checker”"

_pharmaceutics, 2021, doi:10.3390/pharmaceutics13081136_

Round 1
Reviewer 1 Report
The authors describe the development of an in vitro dissolution-permeation system, designed to screen for potential bioequivalence issues. The proposed setup seems to be versatile and can yield interesting results also from mechanistic viewpoint. The manuscript can be considered for publication after the following remarks are addressed:
Materials:
- What was the strength (mg) of the tablets used in the experiments? This information should also be included in the text in section 3.2.
Section 2.2.
Line 119-121: what is this "base medium of FaSSIF", which the authors refer to? None of the components listed here (NaHCO3, glucose, ...) are seen in the composition of the fluids in the donor compartment, described in Table 1.m The authors should described the media composition in a consistent and clear way.
Was an emulsion formed in the donor or the acceptor compartment, due to the stirring and the proximity of the octanol and water phases? Did the octanol phase remain clear throughout the timeframe of all experiments?
Lines 204-206: is the difference in the permeation rate between 200 rpm and 50 or 100 rpm statistically significant? The effect seems to be in the range of the experimental error. Furthermore, the difference in drug release is in the first 30 min of the experiment (where permeation is the same). I suggest the the authors reconsider the interpretation of these results.
General:
What were the drug concentrations measured in the acceptor and the measured permeation rate? Is the permeation part of the method applicable at all for class 2 drugs with very low solubility?
The mismatch between the drug release rate (expected to be similar to in vivo) and the permeation rate (much slower, compared to in vivo) should be mentioned as a disclaimer for the method.
Author Response
to Reviewer 1
The authors describe the development of an in vitro dissolution-permeation system, designed to screen for potential bioequivalence issues. The proposed setup seems to be versatile and can yield interesting results also from mechanistic viewpoint. The manuscript can be considered for publication after the following remarks are addressed:
Thank you very much for your positive and valuable comments on our manuscript. Our responses to your comments are as follows.
Materials:
- What was the strength (mg) of the tablets used in the experiments? This information should also be included in the text in section 3.2.
We have added the strength (mg) to all tablets in the Section 2.1.
Section 2.2.
Line 119-121: what is this "base medium of FaSSIF", which the authors refer to? None of the components listed here (NaHCO3, glucose, ...) are seen in the composition of the fluids in the donor compartment, described in Table 1.m The authors should described the media composition in a consistent and clear way.
Thank you for the indication. We recognized that the description about medium compositions was not appropriate.
We have improved Table 1 and some sentences in Section 2.2 to make it clear.
Was an emulsion formed in the donor or the acceptor compartment, due to the stirring and the proximity of the octanol and water phases? Did the octanol phase remain clear throughout the timeframe of all experiments?
Throughout the experiment, both donor (FaSSIF) and acceptor (octanol) fluids remained clear and formation of emulsions was not observed in both sides. This is because the hydrophilic filter (Durapore filter with 0.22 µm pore) efficiently prevented the migration of both fluids even under the stirring condition. However, when using the filter with larger pore size (0.45 µm), small droplets of aqueous fluid were occasionally observed in the octanol phase due to the leak of donor fluid into the acceptor side. Therefore, we have used the filter with small pore size in this study.
Lines 204-206: is the difference in the permeation rate between 200 rpm and 50 or 100 rpm statistically significant? The effect seems to be in the range of the experimental error. Furthermore, the difference in drug release is in the first 30 min of the experiment (where permeation is the same). I suggest the authors reconsider the interpretation of these results.
Thank you for the comment. Yes, the differences in the permeated amount between 200 rpm and 50 or 100 rpm were not significant and within the range of experimental variations.
Therefore, we have reconsidered the interpretation of the result and added following sentences in the text (Page 6).
“Effect of paddle speed on the permeation profile of metoprolol was not clearly observed despite its pronounced effect on the early phase-profile of the dissolution. Since the filter was not soaked with the donor and receiver fluids at the beginning and the fluid level gradually raised to initiate the permeation of the drug, early-phase differences in the dissolution profile may not be completely reflected to the permeation profile. This is the similar situation with in vivo, because the drug dissolution profile in the stomach just after drug administration may not profoundly affect the absorption profile from the intestine.”
General:
What were the drug concentrations measured in the acceptor and the measured permeation rate? Is the permeation part of the method applicable at all for class 2 drugs with very low solubility?
Since both the donor and acceptor volume changed with time for first 10 or 20 minutes, drug concentration profile seems not informative. Therefore, we have presented the dissolved and permeated amounts of drugs as % of dose in all figures. If necessary, drug permeation rate can be calculated from the slope of the time-course of permeated amount after the infusion of octanol to acceptor chamber was completed (volume became constant).
In this system, dissolved drugs in the donor side permeated the filter and partitioned to octanol which reflected the dissolution in the GI tract and absorption into blood circulation in vivo, the method is applicable for drugs in all BCS classes.
The mismatch between the drug release rate (expected to be similar to in vivo) and the permeation rate (much slower, compared to in vivo) should be mentioned as a disclaimer for the method.
According to the reviewer’s comment, we have added following sentences to explain this in the text (lines 401-412).
“As with the case of D/P system, the drug permeated amount (% of dose) in the BE checker was much smaller than the in vivo absorption (Fa). This is simply due to the lower permeation (absorption) clearance of the filter/octanol system in the BE checker than that in the in vivo absorption from the small intestine, because of the smaller membrane surface area/volume ratio (A/V ratio) in this system. To evaluate the Fa from the data of BE checker, the same method with that used for D/P system is applicable in which the correlation between permeated amount and Fa is used as a standard curve. Also, it is possible to convert the drug permeation profile in the BE checker to its plasma concentration profile after oral administration using in silico PK modeling. Then the obtained parameters such as AUC and Cmax will be used to predict the BE of the test formulation. This is the next project to make the BE checker more useful for BE analysis using in vitro-in silico-in vivo extrapolation (IVISIVE). The study is currently in progress and will soon be reported.”
Reviewer 2 Report
This is an interesting concept bringing together two in vitro models to identify potential bioequivalence issues in some populations including those with achlorhydria and/or weaker gastric agitation. The authors have addressed the concern I would always raise with such efforts in that are we trying to solve a problem that does not clinically exist? Is the test too sensitive and prone to false positives? That said, it is a well-designed and argued study.
Line 50, consider using the term innovator product
It would be helpful to have a few sentences in the introduction justifying the choice of drug and formulations. The Seloken is a very low dose product and it is hard to retrieve much information about it,.
Page 6. The permeation only appears to perhaps change after 90 minutes. What statistical tests were used?
Again, I am not sure that F2 analysis is the best way to determine similarity for permeation studies. Would it be better to look at flux?
There are some grammatical errors that do need addressing throughout the submission
Author Response
to Reviewer 2
This is an interesting concept bringing together two in vitro models to identify potential bioequivalence issues in some populations including those with achlorhydria and/or weaker gastric agitation. The authors have addressed the concern I would always raise with such efforts in that are we trying to solve a problem that does not clinically exist? Is the test too sensitive and prone to false positives? That said, it is a well-designed and argued study.
Thank you very much for your positive and valuable comments on our manuscript. Yes, we understand your saying that our concerns may not clinically exist, and our system may be too sensitive and possible to false positive, because no clear evidences are reported so far about this. However, vice versa, nobody can say that BE product is always BE for all patients. One of the final goals of this study is to make clear this point and to help the development of more “robust products” those can show a stable performance in vivo, and less influenced by the physiological conditions in the GI tract.
Line 50, consider using the term innovator product
Thank you for the indication. I have changed the term to “original product”
It would be helpful to have a few sentences in the introduction justifying the choice of drug and formulations. The Seloken is a very low dose product and it is hard to retrieve much information about it,
As was noted in the Section 3.2, Seloken Tablets 20 mg was used in this study to check the effect of paddle speed because metoprolol dissolves completely and the dissolution profile is not affected by the fluid pH. Therefore, we believe that the effect of stirring strength can be detected simply from the dissolution profile of metoprolol.
Page 6. The permeation only appears to perhaps change after 90 minutes. What statistical tests were used?
Thank you for the comment. The differences in the permeated amount between 200 rpm and 50 or 100 rpm were not significant and within the range of experimental variations.
Therefore, we have reconsidered the interpretation of the result and added following sentences in the text (Page 6).
“Effect of paddle speed on the permeation profile of metoprolol was not clearly observed despite its pronounced effect on the early phase-profile of the dissolution. Since the filter was not soaked with the donor and receiver fluids at the beginning and the fluid level gradually raised to initiate the permeation of the drug, early-phase differences in the dissolution profile may not be completely reflected to the permeation profile. This is the similar situation with in vivo, because the drug dissolution profile in the stomach just after drug administration may not profoundly affect the absorption profile from the intestine.”
Again, I am not sure that F2 analysis is the best way to determine similarity for permeation studies. Would it be better to look at flux?
We fully agree with the reviewer’s comment. F2 analysis was used in this study as one of the possible and easy ways to check quantitatively the similarity or dissimilarity in the permeation profiles of 2 drug formulations. Therefore, it may not be the best method to determine the similarity, however, from the view point of BE, statistical analysis such as t-test may not be appropriate.
We have added following sentences to explain this in the text (lines 401-412).
“As the next study, we are now trying to convert the permeation profile in the BE checker into the plasma-concentration profile of the drug using in silico PK modeling. Then, the obtained parameters such as AUC and Cmax will be used to predict the BE of test formulations. This is the next project to make the BE checker more useful for BE analysis using in vitro-in silico-in vivo extrapolation (IVISIVE). The study is currently in progress and will soon be reported.”
There are some grammatical errors that do need addressing throughout the submission
We have checked the English usage as much as we can and revise them if needed.
Reviewer 3 Report
The paper “Bioequivalence of oral drug products in the healthy and special populations: assessment and prediction using a newly developed in vitro system “BE checker”” describes a new system design to predict bioequivalence (BE) of oral drug products in the healthy and achlorhydria patient populations to develop more biorelevant in vitro system (in vitro bioequivalence checking system (BE checker)) compared to the previously developed SIFC system. Very interesting results were enough for presenting the new system’s outputs and outcomes which are different and valuable than the previously designed SIFC system. Merits to be published.
Author Response
to Reviewer 3
The paper “Bioequivalence of oral drug products in the healthy and special populations: assessment and prediction using a newly developed in vitro system “BE checker”” describes a new system design to predict bioequivalence (BE) of oral drug products in the healthy and achlorhydria patient populations to develop more biorelevant in vitro system (in vitro bioequivalence checking system (BE checker)) compared to the previously developed SIFC system. Very interesting results were enough for presenting the new system’s outputs and outcomes which are different and valuable than the previously designed SIFC system. Merits to be published.
Thank you very much for your positive comments on our manuscript.
Reviewer 4 Report
The paper concerns a biopharmaceutical relevant apparatus for the assessment of dissolution and permeation. The system is interesting for the versatile application to poorly soluble drugs formulations. Different cases of applications are described.The paper is well written and both methods and results are clearly presented.
only a minor observation: in the introduction previous/ analogous in vitro systems are referred with references 4-13. Maybe just a few more lines briefly specifying the characteristics of these systems could be useful for the reader.
Author Response
to Reviewer 4
The paper concerns a biopharmaceutical relevant apparatus for the assessment of dissolution and permeation. The system is interesting for the versatile application to poorly soluble drugs formulations. Different cases of applications are described. The paper is well written and both methods and results are clearly presented. only a minor observation: in the introduction previous/ analogous in vitro systems are referred with references 4-13. Maybe just a few more lines briefly specifying the characteristics of these systems could be useful for the reader.
Thank you very much for your positive comments on our manuscript.
According to the reviewer’s comment, we have added the explanations of those systems briefly (line 71) and rearranged the order of the references.
“As one of those systems, multicompartmental in vitro dissolution apparatus, Gastrointestinal Simulator (GIS), was shown to be able to detect the effect of gastric pH on the dissolution profile of dipyridamole, a BCS class II weak base drug, which corresponded well with the clinically observed drug-drug interaction with famotidine [12]. Additional function of drug partitioning to the octanol phase was also introduced to GIS to observe the process of membrane permeation [13].”
Round 2
Reviewer 1 Report
The authors have addressed my comments.